
# Relating non-Hermitian and Hermitian quantum systems at criticality

**Chang-Tse Hsieh[1] and Po-Yao Chang[2*]**

**1** Department of Physics, National Taiwan University, Taipei 10617, Taiwan
**2** Department of Physics, National Tsing Hua University, Hsinchu 30013, Taiwan

⋆ pychang@phys.nthu.edu.tw

## Abstract

We demonstrate three types of transformations that establish connections between Hermitian and non-Hermitian quantum systems at criticality, which can be described by conformal field theories (CFTs). For the transformation preserving both the energy and the entanglement spectra, the corresponding central charges obtained from the logarithmic scaling of the entanglement entropy are identical for both Hermitian and non-Hermitian systems. The second transformation, while preserving the energy spectrum, does not perserve the entanglement spectrum. This leads to different entanglement entropy scalings and results in different central charges for the two types of systems. We demonstrate this transformation using the dilation method applied to the free fermion case. Through this method, we show that a non-Hermitian system with central charge $c = -4$ can be mapped to a Hermitian system with central charge $c = 2$. Lastly, we investigate the Galois conjugation in the Fibonacci model with the parameter $\phi \to -1/\phi$, in which the transformation does not preserve both energy and entanglement spectra. We demonstrate the Fibonacci model and its Galois conjugation relate the tricritical Ising model/3-state Potts model and the Lee-Yang model with negative central charges from the scaling property of the entanglement entropy.

# 1  Introduction

Non-Hermitian physics has been extensively studied over the past few years in optics [1–6], open quantum systems [7, 8], ultracold atoms [9], and magnetic systems [10–12]. Many interesting extensions from the Hermitian counterparts [13] as well as unique properties due to non-Hermiticity such as the non-Hermitian skin effect [14–16] and the exceptional points [17] were discovered and investigated. On the other hand, the determination of many-body phases and their transitions in non-Hermitian systems was less explored and remains a challenge. A well-known study of the critical behaviors in such systems is the Lee-Yang edge singularity [18–21], which occurs in the exotic phase transition of an Ising model with an imaginary field [22]. The critical point of this transition has been identified as a non-unitary conformal minimal model with a negative central charge [21, 23].

Although the conformal field theory (CFT) description of critical theories can be extended from Hermitian to non-Hermitian systems [24–27], the physical interpretation of negative central charges is still unclear. To have a further insight, constructing transformations that connect the unitary and non-unitary CFTs may provide a deep understanding of the critical behaviors in non-Hermitian systems. In Ref. [28], Guruswamy and Ludwig constructed a "canonical" mapping between $c = -1$ bosonic system ($c = -2$ fermionic system) and $c = 2$ complex scalar theory ($c = 1$ Dirac theory). The construction of this mapping relies on its energy spectrum preserving property in both theories.

In this paper, we further extend this analysis by using the universal scaling property of the entanglement entropy in $(1 + 1)$-dimensional systems at criticality and by considering three types of transformations that relate Hermitian and non-Hermitian systems. The first type of transformation preserves both the energy and the entanglement spectra. This is similar to the duality transformations on the type-I models discussed in Ref. [29]. The identical entanglement spectra imply the central charges in both non-Hermitian and Hermitian systems are the same, and we expect the critical behaviors for the non-Hermitian systems are identical to their Hermitian counterparts. Here the central charge is extracted from the scaling form of the entanglement entropy in Eq. (2). The second type of transformation preserves only the energy spectrum but not the entanglement spectrum. As discussed in Ref. [29], the duality transformations of type-II models are invariably non-Hermitian. In order to construct a transformation that connects a non-Hermitian system to a Hermitian system with an identical energy spectrum, we consider Naimark's dilation [30–36] which can be thought of as embedding a non-Hermitian system in an enlarged Hermitian system with ancilla states [30, 35, 37, 38]. We find that, while the ground state of the non-Hermitian quantum system can have a negative entanglement entropy with logarithmic scaling, the dilated state will always have the positive entanglement entropy. This mapping is demonstrated by the pseudo-Hermitian free fermion model, where we show the central charge $c = -4$ is mapped to $c = 2$. We expect the uncommon phenomena in these non-Hermitian systems at criticality such as negative central charges can be understood through post-selective measurements [37]. Lastly, we consider the transformation which does not preserve both the energy and the entanglement spectra. We demonstrate this transformation by the Galois conjugation in the Fibonacci anyon chain [39, 40]. We show

that Galois conjugation transforms the conformal vacuum in a unitary CFT to the conformal vacuum in a non-unitary CFT. The central charges extracted from the entanglement entropies of the corresponding conformal vacua agree with Ref. [40], where the tricritical Ising model (3-state Potts model) is transformed to the antiferromagnetic (ferromagnetic) Lee-Yang model with a negative central charge.

## 2 Entanglement entropy in non-Hermitian quantum systems at criticality

In a non-Hermitian system $H \neq H^\dagger$, if one assumes that the eigenvalues $\{E_n\}$ of $H$ are not degenerate, then the eigenstates form a bi-orthonormal basis with $H|\psi_{Rn}\rangle = E_n|\psi_{Rn}\rangle$, $H^\dagger|\psi_{Ln}\rangle = E_n^*|\psi_{Ln}\rangle$, $\langle \psi_{Ln}|\psi_{Rm}\rangle = \delta_{n,m}$ [41]. One should note that the existence of an exceptional point violates the non-degeneracy assumption, and the completeness of the bi-orthonormal basis is not maintained. By using the bi-orthonormal formalism, the observables with respect to a given state $|\psi_{R(L)n}\rangle$ can be evaluated by $\langle O \rangle_n = \langle \psi_{Ln}|\hat{O}|\psi_{Rn}\rangle$. Furthermore, one can define the left-right density matrix for a state $|\psi_{R(L)n}\rangle$ as $\rho_n = |\psi_{Rn}\rangle\langle \psi_{Ln}|$. The observable can be written as $\langle O \rangle_n = \mathrm{Tr}[\rho_n \hat{O}]$. If $\hat{O}_A$ is a local operator in subsystem $A$, the corresponding local observable is $\langle \hat{O}_A \rangle = \mathrm{Tr}[\rho_n \hat{O}_A] = \mathrm{Tr}_A[\rho_{An} \hat{O}_A]$. Here $\rho_{An} = \mathrm{Tr}_B \rho_n$ is the reduced density matrix with respect to state $|\psi_{R(L)n}\rangle$. The bi-orthonormal formalism of the observables evaluated by the left-right density matrix is related to the metric operator $g = \sum_n |\psi_{Rn}\rangle\langle \psi_{Ln}|$ that ensures the inner product is positive-definite, *if and only if* the spectrum of the Hamiltonian is real [32, 42, 43]. The positive-definite property of the inner product offers a measurement postulate that is analogous to the Born rule in Hermitian quantum mechanics. A generic entanglement entropy can then be defined using the reduced density matrix defined from the left-right density matrix [44]:

$$S_A := -\mathrm{Tr}[\rho_A \ln |\rho_A|] = -\sum_\alpha \omega_\alpha \ln |\omega_\alpha|. \tag{1}$$

Here $\{\omega_\alpha\} = \mathrm{Spec}(\rho_A)$ represents the eigenvalues of $\rho_A$, which we refer to as the entanglement spectrum. The above definitions are reduced to the usual definitions for Hermitian cases, providing direct comparisons for all the cases studied in this paper.

In the following discussions, we will consider three types of transformations that relate non-Hermitian and Hermitian systems:

- Energy and entanglement spectra preserving transformation: the similarity transformation.

- Energy spectrum preserving and entanglement spectrum non-preserving transformation: the Naimark's dilation.

- Energy and entanglement spectra non-preserving transformation: the Galois conjugation.

The entanglement entropy will be an important quantity we use to characterize systems before and after these transformations.

### 2.1 Entanglement entropy in critical systems

In $(1+1)$-dimensional systems, entanglement entropy can serve as a tool for probing certain characteristics of quantum systems. For instance, the entanglement entropy, as a function of

subsystem size, displays a constant value for a gapped ground state. For a critical system where the system has a conformal symmetry, the entanglement entropy exhibits a universal scaling [45–50]

$$
\begin{aligned}
S_A &= \frac{c}{3} \ln L_A + \text{const.}, \quad \text{PBC}, \\
&= \frac{c}{6} \ln L_A + \text{const.}, \quad \text{OBC},
\end{aligned}
\tag{2}
$$

where $P(O)BC$ stands for the periodic (open) boundary condition, and $c$ is the corresponding central charge of the underlying CFT. For a lattice system with size $L$, Eq. (2) can be expressed as $S_A = \frac{c}{3} \ln[\sin[\frac{\pi L_A}{L}]] + \cdots$. One can also compute the entanglement entropy by using equal bipartition and varying the total system size. The resulting entanglement entropy for equal bipartition is denoted as $S_A = \frac{c}{3} \ln L_A + \cdots$. It is worth noting that the entanglement spectrum alone can also be described by a universal scaling function depending on the central charge [51, 52]. For a unitary CFT, the unitarity condition enforces that the central charge must be a positive number. However, if one relaxes the unitarity condition, the system can still exhibit conformal symmetry and the underlying description of the system can be a non-unitary CFT. Without the unitarity condition, the central charge is not required to be positive and in principle can be negative or even complex.

One interesting phenomenon that can be described by a non-unitary CFT is percolation, where the corresponding central charge is $c = 0$ [53]. Another notable phenomenon is the Yang-Lee edge singularity [18–20], which describes the zeros of the partition function on the complex external magnetic field $h$ of the classical Ising model. Alternatively, the Yang-Lee edge singularity can be realized as a $(1 + 1)$-dimensional non-Hermitian quantum Hamiltonian of a transversed Ising model with an imaginary longitudinal field. This model, at criticality, is identified as a non-unitary CFT with the central charge $c = -22/5$ [21, 22]. The $bc$-ghost CFTs are other non-Hermitian quantum systems that have been investigated in Refs. [24–26, 54, 55]: $c = -26$ $bc$-ghost CFT appears in worldsheet string theory, and $c = -2$ $bc$-ghost CFT is regarded as the non-logarithmic part of a certain logarithmic CFT with the same central charge [25, 56–58].

It has been shown that Eq. (2) can extract the central charge for many Hermitian quantum systems [45–50]. Recently, the universal scaling property of entanglement entropy at criticality has been extended to non-Hermitian quantum systems, and the corresponding negative central charge can also be extracted [27, 44, 59, 60]. In the following, we demonstrate three relations which connect the Hermitian and non-Hermitian quantum systems at criticality and extract the central charge from the universal scaling property of the entanglement entropy in Eq. (2). For simplicity, we only consider the entanglement entropy scaling for the periodic boundary condition.

A subtlety can arise when the systems exhibit exceptional points. Due to the coalescence of eigenstates, the *generic* entanglement entropy can diverge. In principle, the critical points have properties that differ from those of the exceptional points in terms of fidelity and fidelity susceptibility [61, 62]. However, in the free fermion case studied in Refs. [27, 44] and the second case in this paper, the critical point identified by the gap closing point of a specific momentum $k$ is composed of two exceptional points. For this scenario, we avoid the singularity by introducing a momentum shift $\delta \to 0$ and compute the *generic* entanglement entropy.

## 2.2 Energy and entanglement spectra preserving mapping — similarity transformation

Our example in the first case is the non-Hermitian Ising model introduced in Ref. [63],

$$H = -\sum_{i=1}^{N} \left( J\sigma_i^z \sigma_{i+1}^z + \epsilon_1 \sigma_i^+ + \epsilon_2 \sigma_i^- \right), \tag{3}$$

where $\sigma_i^{\pm} = \sigma_i^x \pm i\sigma_i^y$ with $\sigma_i^{\alpha=x,y,z}$ being the Pauli matrices, and $\epsilon_{1,2}$ are complex numbers with $\epsilon_1 \neq \epsilon_2^*$. The mapping that preserves both the energy and the entanglement spectra of this model is achieved by a similarity transformation, which can be performed by the following operator:

$$\rho = \prod_i \rho_i, \qquad \rho_i = \gamma^{-1/2}\sigma_i^+ \sigma_i^- + \gamma^{1/2}\sigma_i^- \sigma_i^+, \tag{4}$$

where $\gamma = \sqrt{|\epsilon_1|/|\epsilon_2|}$. We can use the identities $\rho\sigma_i^z\rho^{-1} = \sigma_i^z$, $\rho\sigma_i^{\pm}\rho^{-1} = \gamma^{\mp 1}\sigma_i^{\pm}$, and $[\rho_i, \rho_j] = 0$ to show that

$$h = \rho H \rho^{-1} = -\sum_{i=1}^{N} J\sigma_i^z \sigma_{i+1}^z - \beta \left( e^{i\arg(\epsilon_1)}\sigma_i^+ + e^{i\arg(\epsilon_2)}\sigma_i^- \right), \tag{5}$$

where $\beta := \sqrt{|\epsilon_1||\epsilon_2|}$. The quasi-Hermitian condition is held when $\arg(\epsilon_1) + \arg(\epsilon_2) = 2k\pi$, $k \in \mathbb{Z}$. In this situation, $h$ is Hermitian and can be further mapped to the Hermitian transverse Ising model using an unitary transformation. The latter becomes critical and has central charge $c = 1/2$ when $J/\beta = 1$. Note that $H$ is non-Hermitian for $\epsilon_1 \neq \epsilon_2^*$, but the spectrum is real under this quasi-Hermitian condition. As the spectrum is preserved under the similarity transformation (5), the system also remains critical when $J/\beta = 1$, even when $H$ is not Hermitian. Moreover, as shown in Fig. 1(a), the entanglement spectra of the ground states in both Hermitian and non-Hermitian models are identical. Therefore, the entanglement entropy of this quasi-Hermitian model with $J/\beta = 1$ also has $c = 1/2$, as depicted in Fig. 1(b). One can further apply the Jordan-Wigner transformation to map the spin Hamiltonian to a fermionic Hamiltonian. For the non-Hermitian case when $\epsilon_1 \neq \epsilon_2^*$, the mapped fermionic Hamiltonian becomes non-local. However, the (right) ground state can be obtained by the similarity transformation $|R\rangle = \rho^{-1}|\psi\rangle$, where $|\psi\rangle$ represents the ground state of the Hermitian Ising model. The fermionic Hamiltonian of the Hermitian Ising model is local and is bi-linear in fermion operators, in which the entanglement entropy can be computed by the correlation matrix method [64]; the result shown in Fig. 1(c).

## 2.3 Energy spectrum preserving and entanglement spectrum non-preserving mapping — Naimark's dilation

In the second case, we consider a non-Hermitian system with real energy eigenvalues, but the eigenvalues of the reduced density matrix constructed from the bi-orthogonal basis are not necessary real. In order to map this non-Hermitian system to a Hermitian system that the mapped reduced density matrix has real eigenvalues and preserve the energy spectrum, we consider the Naimark's dilation [30, 38].

This dilation method is described in the following steps. First, let us suppose a Hermitian Hamiltonian $H$ has the form

$$H = \begin{pmatrix} H_1 & H_2 \\ H_2^{\dagger} & H_4 \end{pmatrix}, \qquad H_{1(4)}^{\dagger} = H_{1(4)}. \tag{6}$$

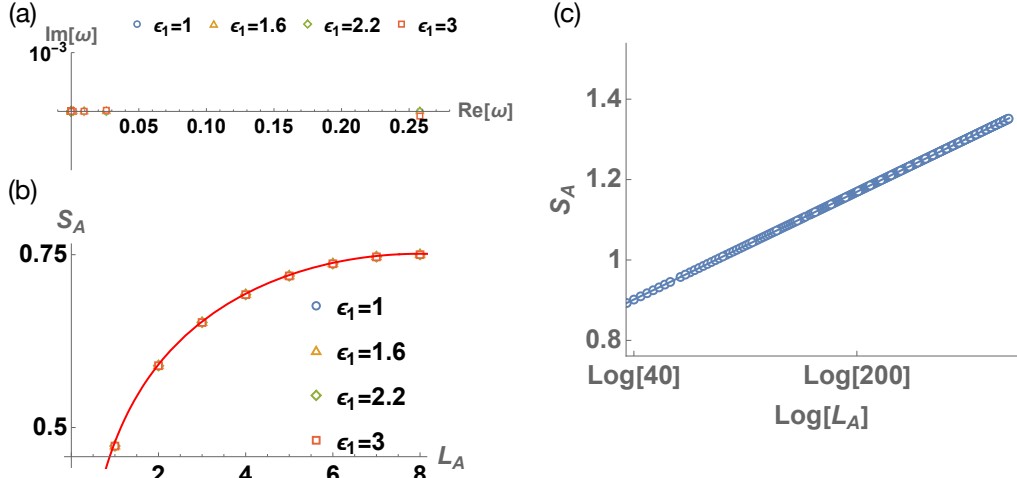

Figure 1: (a) The entanglement spectrum for different parameters $\epsilon_2 = 1/\epsilon_1$ with $\epsilon_1 = 1, 1.6, 2.2, 3$ to ensure that $J/\beta = 1$. (b) The entanglement entropy $S_A$ as a function of the subsystems size $L_A$. Here we set $J = 1$. For both (a) and (b), the total system size is $L = 16$. The scaling of the entanglement entropy as a function of the subsystem $L_A$ is $S_A = 0.48 + 0.166667 \ln[\sin[\frac{\pi L_A}{L}]]$ (red line) (c) The entanglement entropy as a function of the subsystem $L_A$ for equal bipartition $L_A = L/2$ for the transformed Hermitian Ising model at criticality $J/\beta = 1$. The scaling of the entanglement entropy is $S_A = 0.288 + 0.166315 \ln L_A$. Both (b) and (c) give the central charge $c = 1/2$.

We want to find a pseudo-Hermitian $\mathcal{H}$ that shares the same energy spectrum of $H$. We consider the following eigenvalue equations

$$H \begin{pmatrix} \psi \\ \tau\psi \end{pmatrix} = \begin{pmatrix} \mathcal{H}\psi \\ \tau\mathcal{H}\psi \end{pmatrix} = E \begin{pmatrix} \psi \\ \tau\psi \end{pmatrix}, \tag{7}$$

where $\psi$ is the right eigenstate which satisfies $\mathcal{H}\psi = E\psi$. The last step is to solve the matrix equations with $H_1, H_2, H_4$, and $\tau$ that satisfy

$$H_1 + H_2\tau = \mathcal{H}_k, \qquad H_2^\dagger + H_4\tau = \tau\mathcal{H}_k. \tag{8}$$

One should be notified that these matrix equations are not guaranteed to have solutions.

For a concrete example, we consider the pseudo-Hermitian tight-binding model

$$H_{\text{PTB}} = \sum_{i=1}^{N} \frac{1}{2} \left( c_{A,i+1}^\dagger c_{A,i} - c_{A,i}^\dagger c_{A,i+1} - c_{B,i+1}^\dagger c_{B,i} + c_{B,i}^\dagger c_{B,i+1} + c_{A,i}^\dagger c_{B,i} + c_{B,i}^\dagger c_{A,i} \right), \tag{9}$$

where $c_{A(B),i}^{(\dagger)}$ is the fermion operator, $A(B)$ is the site label, $i$ is the unit-cell label. This model can be regarded as the two-leg version of the Hatano-Nelson model [65,66], as shown in Fig. 2(a). For translation invariant situation, the Hamiltonian can be diagonalized in the momentum space $H_{\text{PTB}} = \sum_k c_k^\dagger \mathcal{H}_k c_k$ with $c_k = (c_{A,k}, c_{B,k})$, $k = 2\pi n/N$, $n = 1, \cdots, N$ and

$$\mathcal{H}_k = \begin{pmatrix} i\sin k & 1 \\ 1 & -i\sin k \end{pmatrix}. \tag{10}$$

This model is similar to the two-level system discussed in Ref. [38], with $k$ being a tuning parameter. The pseudo-Hermiticity has the property $\eta\mathcal{H}_k = \mathcal{H}_k^\dagger\eta$ with

$$\eta = \frac{2}{\cos^2 k} \begin{pmatrix} 1 & -i\sin k \\ i\sin k & 1 \end{pmatrix}, \tag{11}$$

which ensures the energy spectrum is real or the complex eigenvalues come in complex conjugate pairs [67]. In this case, the energy spectrum is real, $E_k = \pm \cos k$. There are two crossings with linear dispersion at $k = \pm \pi/2$. The corresponding dilation can be obtained with the following matrices

$$\tau = \frac{1}{\cos k} \begin{pmatrix} 1 & -i \sin k \\ i \sin k & 1 \end{pmatrix},$$

$$H_1 = \frac{\cos k}{2}(\tau \mathcal{H}_k + \mathcal{H}_k \tau^{-1}) = \begin{pmatrix} 0 & \cos^2 k \\ \cos^2 k & 0 \end{pmatrix},$$

$$H_2 = (\mathcal{H}_k - H_1)\tau^{-1} = \begin{pmatrix} i \sin k \cos k & 0 \\ 0 & -i \sin k \cos k \end{pmatrix},$$

$$H_4 = (\tau \mathcal{H}_k - H_2^\dagger)\tau^{-1} = \begin{pmatrix} 0 & \cos^2 k \\ \cos^2 k & 0 \end{pmatrix}. \tag{12}$$

The dilated Hamiltonian is a four by four matrix with the identical energy spectrum $E_k = \pm \cos k$.

The lower (occupied) energy eigenstates of $\mathcal{H}_k$ are

$$|\psi_R^-\rangle = \begin{cases} \frac{1}{\sqrt{1+e^{2ik}}} \begin{pmatrix} 1 \\ -e^{ik} \end{pmatrix}, & |k| \leq \pi/2, \\ \frac{1}{\sqrt{1+e^{-2ik}}} \begin{pmatrix} 1 \\ e^{-ik} \end{pmatrix}, & \pi/2 < |k| \leq \pi. \end{cases} \tag{13}$$

The lower energy eigenstate of $\mathcal{H}_k^\dagger$ are

$$|\psi_L^-\rangle = \begin{cases} \frac{1}{\sqrt{1+e^{-2ik}}} \begin{pmatrix} 1 \\ -e^{-ik} \end{pmatrix}, & |k| \leq \pi/2, \\ \frac{1}{\sqrt{1+e^{2ik}}} \begin{pmatrix} 1 \\ e^{ik} \end{pmatrix}, & \pi/2 < |k| \leq \pi. \end{cases} \tag{14}$$

We can compute the entanglement entropy directly from the single-particle eigenstates using the correlation matrix method [27, 44]. The single-particle entanglement spectrum $\epsilon_E$ can also be obtained from the correlation matrix $\langle \psi_L^- | c_{\alpha,i}^\dagger c_{\beta,j} | \psi_R^- \rangle$ with $\alpha(\beta) = A, B$. Suppose that the eigenvalues of the correlation matrix are $\zeta_\alpha$, the single-particle entanglement spectrum $\epsilon_{E\alpha}$ is then given by $\ln[\zeta_\alpha^{-1} - 1]$ as shown in Fig. 2(b) [orange triangles]. Due to the pseudo-Hermiticity, the single-particle entanglement spectrum has eigenvalues that are either conjugation pairs or real numbers. This property results in a real value for the *generic* entanglement entropy. The entanglement entropy as a function of the total size with equal bipartition $L_A = L/2$ is shown in Fig. 2(c). The fitting perfectly agrees with the CFT prediction $S_A = \frac{c}{3}\ln(L_A) + \cdots$ and the extracted central charge is $c = -4$. One should be notified that the there are two linearly dispersive fermions at $k = \pm \pi/2$. However, these $k$ points are the exceptional points where two eigenstates coalesce, and the expression of the single-particle states in Eqs. (13) and (14) diverges. To avoid this divergence, we compute the entanglement entropy by slightly shifting the momentum at $k = \pm \pi/2$ by a small value $\delta = 0.000001$, as discussed in Ref. [27, 44]. Each linearly dispersive fermion can be described by the *bc*-ghost CFT with $c = -2$ and in total this model has two linearly dispersive fermions which agrees with our numerical observation.

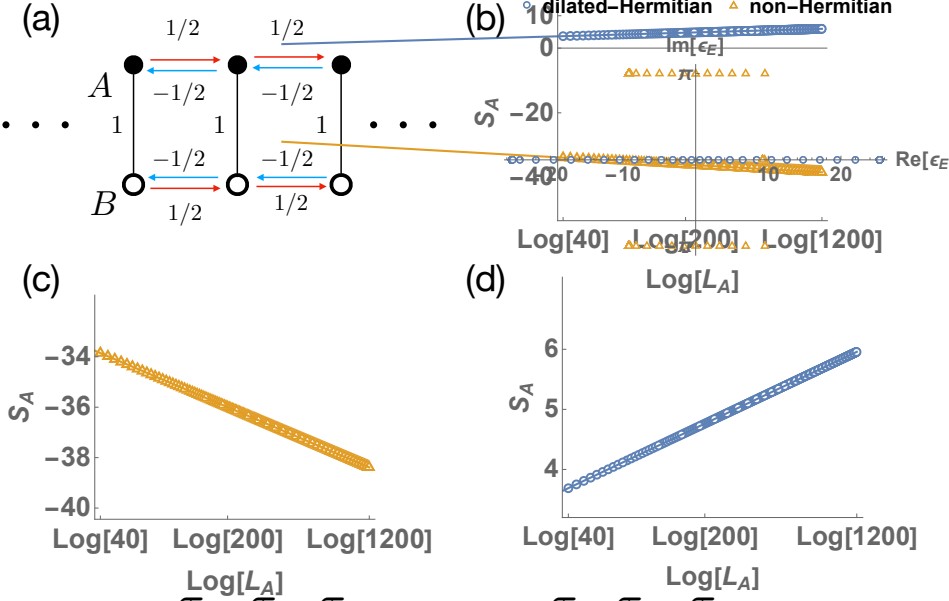

Figure 2: (a) The pseudo-Hermitian tight-binding model described by Eq. (9). The arrows indicate the asymmetric hopping terms, $1/2$ and $-1/2$. (b) The single-particle entanglement spectrum of Eq. (9) (orange triangles) and its dilated one (blue circles). (c) The entanglement entropy of Eq. (9). (d) The entanglement entropy of the dilated system. The entanglement entropy scaling for (c) is $S_A = 1.2289 + 0.6669 \ln L_A$, and for (d) it is $S_A = -28.9344 - 1.333 \ln L_A$, leading to the central charges $c = 2$ and $c = -4$, respectively.

The dilated effective free fermion theory preserves the same spectrum and its lower eigenstates can be directly obtained from the right eigenstate $|\psi_R^-\rangle$ as

$$|\Psi^-\rangle = \begin{pmatrix} |\psi_R^-\rangle \\ \tau|\psi_R^-\rangle \end{pmatrix}. \tag{15}$$

We can compute the entanglement entropy and the single-particle entanglement spectrum from the states $|\Psi^-\rangle$ by the correlation matrix [64]. The single-particle entanglement spectrum is real as shown in Fig. 2(b) [blue circles]. We extract the central charge from the entanglement entropy scaling $S_A = \frac{c}{3}\ln L_A + \cdots$. The extracted central charge is $c = 2$, which can be described by two linear dispersive fermions with $c = 1$ for each fermion [Fig. 2(d)]. The mapping from $c = -2$ to $c = 1$ has been discussed in Ref. [28] from comparison of the partition functions of $bc$-ghost fermion and the Dirac fermion. By using the Naimark's dilation method, the central charges extracted from the ground states of both Hermitian and non-Hermitian systems agree with the partition function analysis [28].

## 2.4 Energy and entanglement spectra non-preserving mapping — Galois conjugation

In the last case where both the energy and the entanglement spectra are not preserved, we consider the Fibonacci anyon chain introduced in Ref. [39, 40], where the Hamiltonian $H = \sum_{i=1}^N H_i$ is constructed from the local fusion rules of two neighboring anyons. We focus on the Fibonacci anyons where the fusion rules are $\tau \times \tau = 1 + \tau$, $1 \times \tau = \tau \times 1 = \tau$, and $1 \times 1 = 1$. The Hamiltonian is designed by assigning $E_\tau = 0$ for $x_i' = \tau$ (two anyons fuse to vacuum) and $E_1 = -1$ for $x_i' = 1$ (two anyons fuse to $\tau$). Since the local anyon ($\tau$) basis and

Figure 3: The change of the link basis by F-matrix.

the link basis can be transformed by the $F$-matrix, one can express the local terms $H_i$ in terms of the three-body interactions in the link basis as shown in Fig. 3,

$$H_i|x_{i-1}x_ix_{i+1}\rangle = \sum_{x_i'=1,\tau} (H_i)_{x_i}^{x_i'}|x_{i-1}x_i'x_{i+1}\rangle, \qquad (H_i)_{x_i}^{x_i'} := -(F_{x_{i-1}\tau\tau}^{x_{i+1}})_{x_i}^1 (F_{x_{i-1}\tau\tau}^{x_{i+1}})_{x_i'}^1. \tag{16}$$

For other matrix elements $\{|x_{i-1}x_ix_{i+1}\rangle\} = \{|1\tau1\rangle, |1\tau\tau\rangle, |\tau\tau1\rangle\}$, the local Hamiltonian is diagonal $H_i = \mathrm{diag}\{-1,0,0\}$. For $x_{i-1} = x_{i+1} = \tau$, the $F$-matrix is

$$F_{\tau\tau\tau}^{\tau} = \begin{pmatrix} \phi^{-1} & \phi^{-1/2} \\ \phi^{-1/2} & -\phi^{-1} \end{pmatrix}, \tag{17}$$

and the corresponding $H_i$ is

$$H_i = -\begin{pmatrix} \phi^{-2} & \phi^{-3/2} \\ \phi^{-3/2} & \phi^{-1} \end{pmatrix}. \tag{18}$$

Here $\phi = \frac{1}{2}(1+\sqrt{5})$. One can further express the local Hamiltonian in terms of the Pauli matrices

$$H_i = (n_{i+1} + n_{i-1} - 1) - n_{i+1}n_{i-1}\left(\phi^{-3/2}\sigma_i^x + \phi^{-3}n_i + 1 + \phi^{-2}\right), \tag{19}$$

where $n_i = (1 - \sigma_i^z)/2 = 0,1$ being the counting of $\tau$ particle on the $i$th-link and the auxiliary basis $\{|111\rangle, |11\tau\rangle, |\tau11\rangle\}$ needs to be projected out. As pointed out in Ref. [39], this model with an antiferromagnetic coupling (energetically favoring the fusion to 1 channel) gives $c = 7/10$ (tricritical Ising model), while the same model with a ferromagnetic coupling (energetically favoring the fusion to $\tau$ channel) gives $c = 4/5$ (3-state Potts model).

From the algebraic point of view, the construction of the Fibonacci anyon chain is based on the fusion rule $\tau \times \tau = 1 + \tau$ which can be described by $x^2 = 1 + x$. The solution is the golden ration $x = \frac{1}{2}(1+\sqrt{5})$. The process of the Galois conjugation is taking $\phi \to -1/\phi$ which is the other solution of $x^2 = 1 + x$. It is shown in Ref. [40] that the antiferromagnetic conjugated model has $c = -3/5$ (antiferromagnetic Yang-Lee chain) while the ferromagnetic conjugated model has $c = -22/5$ (ferromagnetic Yang-Lee chain).

To have further insight of the Galois conjugation, we first consider the three-site case where the dimension of the Hamiltonian after the projection is four. The matrix representation of the Hamiltonian (antiferromagnetic coupling) for the basis $\{|1\tau\tau\rangle, |\tau1\tau\rangle, |\tau\tau1\rangle, |\tau\tau\tau\rangle\}$ is

$$H = \begin{pmatrix} 3 - \frac{1}{\phi^2} & 0 & 0 & -\frac{1}{\phi^{3/2}} \\ 0 & 3 - \frac{1}{\phi^2} & 0 & -\frac{1}{\phi^{3/2}} \\ 0 & 0 & 3 - \frac{1}{\phi^2} & -\frac{1}{\phi^{3/2}} \\ -\frac{1}{\phi^{3/2}} & -\frac{1}{\phi^{3/2}} & -\frac{1}{\phi^{3/2}} & 3 - \frac{3}{\phi^2} - \frac{3}{\phi^3} \end{pmatrix}. \tag{20}$$

The corresponding eigenvalues are $E_1 = E_2 = -3 + \frac{1}{\phi^2}$, and $E_\pm = \frac{1}{2\phi^3}(3 + 4\phi - 6\phi^3 \pm \sqrt{9 + 12\phi + 4\phi^2 + 12\phi^3})$. For the Hermitian case $\phi = \frac{1}{2}(1+\sqrt{5})$, the energies are



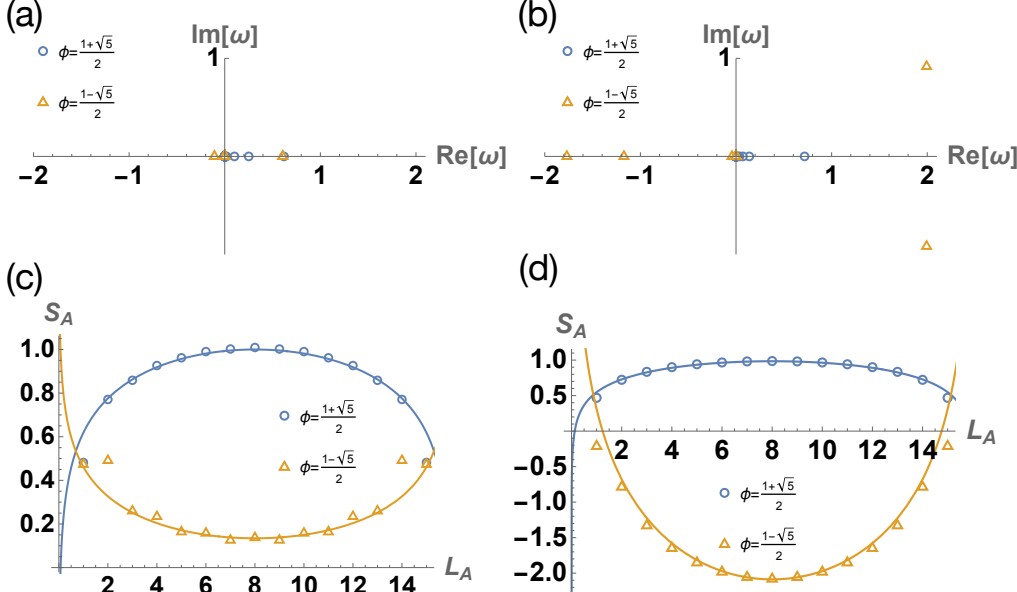

Figure 4: (a)(b): The entanglement spectra for the antiferromagnetic and ferromagnetic cases, respectively, with $(L_A, L) = (8, 16)$. (c)(d): The corresponding entanglement entropies as functions of the subsystem size $L_A$ for the antiferromagnetic and ferromagnetic cases, respectively, with $L = 16$. The solid curves represents the CFT prediction $S_A = c/3 \log[(16 \sin[\pi L_A/16])/\pi) + \text{const.}]$. For the antiferromagnetic case (c), the blue curve corresponds to $c = 7/10$ and the orange curve corresponds to $c = -3/5$. For the ferromagnetic case (d), the blue curve corresponds to $c = 4/5$ and the orange curve corresponds to $c = -22/5$.

$E_1 = E_2 = -2.61803$, $E_+ = -0.763932$, $E_- = -3$. The lowest energy is $E_-$ and the corresponding eigenstate is the ground state. For the non-Hermitian case under Galois conjugation $\phi = \frac{1}{2}(1 - \sqrt{5})$, the energies are $E_1 = E_2 = 0.381966$, $E_+ = -5.23607$, $E_- = -3$, the lowest energy changes to $E_+$ instead of $E_-$, and $E_-$ becomes the first excited energy.

Suppose $|\psi_{E_-}(\phi = \frac{1}{2}(1 + \sqrt{5}))\rangle$ is the ground state in the Hermitian case, which has the lowest energy $E_- = -3$, the Galois conjugated Hamiltonian transform the $|\psi_{E_-}(\phi = \frac{1}{2}(1 + \sqrt{5})\rangle$ to $|\psi_{E_-}(\phi = \frac{1}{2}(1 - \sqrt{5}))\rangle$, which is the first excited state instead of the ground state. In fact, the lowest energy state for the Galois conjugated Hamiltonian is not the conformal vacuum anymore [40]. In this non-unitary CFT, the lowest energy state corresponds to the primary field with negative conformal weight, whereas the conformal vacuum is the first excited state. In the three-site example, we can clearly observe that the Galois conjugation transforms the conformal vacuum (the ground state $|\psi_{E_-}(\phi = \frac{1}{2}(1 + \sqrt{5})\rangle$) in the unitary CFT to the conformal vacuum (the first excited state $|\psi_{E_-}(\phi = \frac{1}{2}(1 - \sqrt{5}))\rangle$) in the non-unitary CFT.

A more sophisticated demonstration of the vacuum preserving mapping from the Galois conjugation can be shown from the matrix product operator (MPO) algebra of string-net projected entangled pair state (PEPS) [68,69]. One can construct the MPO operators $A_{abcd}$ which are the basis elements of a $C^*$ algebra from the string-net PEPS. The central idempotents of these MPO operators are the projectors of different topological sectors. For the Fibonacci string-net ($\phi = \frac{1}{2}(1 + \sqrt{5})$), the sector for conformal vacuum is $\mathcal{P}_{1,11} = \frac{1}{\sqrt{5}}[\frac{1}{\phi}A_{1111} + A_{1\tau1\tau}]$. The Galois conjugation transforms this vacuum sector as $\mathcal{P}_{1,11} = -\frac{1}{\sqrt{5}}[\frac{1}{\phi}A_{1111} + A_{1\tau1\tau}]$ with $\phi = \frac{1}{2}(1 - \sqrt{5})$ which indeed is the vacuum sector of the Yang-Lee model discussed in Refs. [70, 71]. Here, the vacuum sector corresponds to the first excited state with a conformal weight

$h = 0$, while the ground state is associated with a negative conformal weight $h = -1/5$. One can also clarify that $\mathcal{P}_{1,11}^2 = \mathcal{P}_{1,11}$ is the projector from the multiplication $A_{1111}A_{1111} = A_{1111}$, $A_{1111}A_{1\tau1\tau} = A_{1\tau1\tau}A_{1111} = A_{1\tau1\tau}$, and $A_{1\tau1\tau}A_{1\tau1\tau} = A_{1111} + A_{1\tau1\tau}$. The Galois conjugation can be performed for other topological sectors; we discuss these mappings in the appendix A for completeness.

Given this property, we examine both the entanglement spectrum and the generic entanglement entropy of the ground state of the Hermitian golden chain and the same characteristics of the first excited state of the Galois conjugated Hamiltonian. This helps establish a relation between the unitary CFT and non-unitary CFT through their respective conformal vacua. The entanglement spectra for the Hermitian cases are real and fall within the region $\omega_\alpha \in [0,1]$, as shown in the antiferromagnetic case in Fig. 4(a) [blue circles] and the ferromagnetic case in Fig. 4(b) [blue circles]. For the non-Hermitian cases, the entanglement spectra can be complex and outside the region $\omega_\alpha \in [0,1]$, as shown in the antiferromagnetic case in Fig. 4(a) [orange triangles] and the ferromagnetic case in Fig. 4(b) [orange triangles]. However, the complex eigenvalues of the left-right reduced density always come in conjugate pairs, leading to a real *generic* entanglement entropy. The results of the *generic* entanglement entropy are shown for the antiferromagnetic case in Fig. 4(c) and for the ferromagnetic case in Fig. 4(d). The scaling of the entanglement entropy leads to the desired central charges predicted by the CFTs.

## 3 Conclusion

We demonstrate three types of transformations that relate Hermitian and non-Hermitian at criticality. For the transformation that preserves both the energy and the entanglement spectra, such as similarity transformations, the central charges of the critical systems before and after the transformation are the same. A simple example is the quasi-Hermitian Ising model which can be mapped to the Hermitian transverse Ising model, both having $c = 1/2$ at criticality. We also construct the transformation that preserves the energy but not the entanglement spectra using the Naimark's dilation method. The model we consider in this case is the pseudo-Hermitian tight-binding model, which has $c = -4$ and is mapped to a gapless free Dirac fermions with $c = 2$. Lastly, the Galois conjugation in the Fibonacci anyon chain preserves neither the energy nor the entanglement spectra. We show that the Galois conjugation transforms the conformal vacuum in an unitary CFT to the conformal vacuum in a non-unitary CFT, and the corresponding central charges can be correctly extracted from the "generic" entanglement entropy scaling.

## Acknowledgements

PYC would like to thank Ray-Kuang Lee for introducing the Naimark's dilation. The authors thank Pochung Chen, Chong-Sun Chu, Kohei Kawabata, Chung-Yu Mou, Shinsei Ryu, and Ken Shiozaki for valuable discussions. The authors are also grateful for support from National Center for Theoretical Sciences in Taiwan.

**Funding information** PYC is supported by the Young Scholar Fellowship Program by National Science and Technology Council (NSTC) in Taiwan, under grant No.112-2636-M-007-007. CTH is supported by the Yushan (Young) Scholar Program of the Ministry of Education in Taiwan under grant NTU-111VV016.

# A  Fibonacci string-net model

The topological data of the Fibonacci string-net from the fusion rules are $N_{11}^1 = N_{\tau\tau}^1 = N_{\tau\tau}^\tau = N_{1\tau}^\tau = N_{\tau 1}^\tau = 1$, and others are zero. Here $N_{\alpha\beta}^\gamma$ is defined from the fusion rules $\alpha \otimes \beta = \oplus_\gamma N_{\alpha\beta}^\gamma \gamma$. We can get the non-trivial elements of $F$ from the pentagon equations,

$$[F_{\tau\tau\tau}^\tau]_{d,a}[F_{a\tau\tau}^\tau]_{c,b} = \sum_e [F_{\tau\tau\tau}^d]_{c,e}[F_{\tau e\tau}^\tau]_{d,b}[F_{\tau\tau\tau}^b]_{e,a}, \tag{A.1}$$

which leads to

$$[F_{\tau\tau\tau}^\tau]_{a,b} = \begin{pmatrix} \phi^{-1} & \phi^{-1/2} \\ \phi^{-1/2} & -\phi^{-1} \end{pmatrix}, \tag{A.2}$$

with $\phi = \frac{1}{2}(1+\sqrt{5})$.

## A.1  Ocneanu's tube algebra

Ocneanu's tube algebra is a classification scheme of quasi-particles in the string-net models. The basic idea is one can view a quasiparticle is a local area with higher energy density surrounded by the ground state area. Since any local operations acting on the area of the quasiparticle cannot change the types of quasiparticle and the quasiparticle is scale invariant, we can glue another ground state area (annulus geometry) on the original system without changing the types of quasiparticle. Mathematically, it suggests for the ground states on a cylinder which is a subspace $\mathcal{V}_{\text{cyl}}$ of the total Hilbert space, gluing two cylinders along one boundary also forms a subspace $\mathcal{V}_{\text{cyl}}$ of ground states, which can be expressed as

$$\mathcal{V}_{\text{cyl}} \otimes \mathcal{V}_{\text{cyl}} \to \mathcal{V}_{\text{cyl}}. \tag{A.3}$$

This mapping forms an algebra. In the string-net projected entanglement pair state (PEPS), the ground states on a cylinder can be expressed in terms of the matrix product operators (MPOs) as shown in Fig. 5 (a). These MPO representations of the ground states on a cylinder forms the Ocneanu's tube algebra [Fig. 5 (b)]

$$A_{abcd}A_{cefg} = \sum_{h,i} C_{(abcd),(cefg)}^{ahfi} A_{ahfi}. \tag{A.4}$$

This tube algebra can be decomposed by a direct sum of central idempotents, which we denote as $\mathcal{P}_i$. These central idempotents satisfy $\mathcal{P}_i \mathcal{P}_j = \delta_{ij}\mathcal{P}_j$ and $A_{abcd}\mathcal{P}_i = \mathcal{P}_i A_{abcd}$.

In the Fibonacci string-net model, the tube-algebra is seven-dimensional with the basis $\{A_{1111}, A_{1\tau 1\tau}, A_{\tau\tau 1}, A_{\tau 1\tau\tau}, A_{\tau\tau\tau\tau}, A_{1\tau\tau\tau}, A_{\tau\tau 1\tau}\}$. The multiplication table is

|  | $A_{1111}$ | $A_{1\tau 1\tau}$ | $A_{\tau\tau 1}$ | $A_{\tau 1\tau\tau}$ | $A_{\tau\tau\tau\tau}$ | $A_{1\tau\tau\tau}$ | $A_{\tau\tau 1\tau}$ |
|---|---|---|---|---|---|---|---|
| $A_{1111}$ | $A_{1111}$ | $A_{1\tau 1\tau}$ | 0 | 0 | 0 | $A_{1\tau\tau\tau}$ | 0 |
| $A_{1\tau 1\tau}$ | $A_{1\tau 1\tau}$ | $A_{1111}+A_{1\tau 1\tau}$ | 0 | 0 | 0 | $-\frac{A_{1\tau\tau\tau}}{\phi}$ | 0 |
| $A_{\tau\tau 1}$ | 0 | 0 | $A_{\tau\tau 1}$ | $A_{\tau 1\tau\tau}$ | $A_{\tau\tau\tau\tau}$ | 0 | $A_{\tau\tau 1\tau}$ |
| $A_{\tau 1\tau\tau}$ | 0 | 0 | $A_{\tau 1\tau\tau}$ | $\frac{A_{\tau\tau 1}}{\phi}+\frac{A_{\tau\tau\tau\tau}}{\sqrt{\phi}}$ | $\frac{A_{\tau\tau 1}}{\sqrt{\phi}}-\frac{A_{\tau\tau\tau\tau}}{\phi}$ | 0 | $A_{\tau\tau 1\tau}$ |
| $A_{\tau\tau\tau\tau}$ | 0 | 0 | $A_{\tau\tau\tau\tau}$ | $\frac{A_{\tau\tau 1}}{\sqrt{\phi}}-\frac{A_{\tau\tau\tau\tau}}{\phi}$ | $-\frac{A_{\tau\tau 1}}{\phi}+A_{\tau 1\tau\tau}-\frac{A_{\tau\tau\tau\tau}}{\phi^2\sqrt{\phi}}$ | 0 | $\frac{A_{\tau\tau 1\tau}}{\phi\sqrt{\phi}}$ |
| $A_{1\tau\tau\tau}$ | 0 | 0 | $A_{1\tau\tau\tau}$ | $A_{1\tau\tau\tau}$ | $\frac{A_{1\tau\tau\tau}}{\phi\sqrt{\phi}}$ | 0 | $\sqrt{\phi}A_{1111}-\frac{A_{1\tau 1\tau}}{\sqrt{\phi}}$ |
| $A_{\tau\tau 1\tau}$ | $A_{\tau\tau 1\tau}$ | $-\frac{A_{\tau\tau 1\tau}}{\phi}$ | 0 | 0 | 0 | $\frac{A_{\tau\tau 1}}{\sqrt{\phi}}+\frac{A_{\tau 1\tau\tau}}{\sqrt{\phi}}+\frac{A_{\tau\tau\tau\tau}}{\phi^2}$ | 0 |

$$\tag{A.5}$$

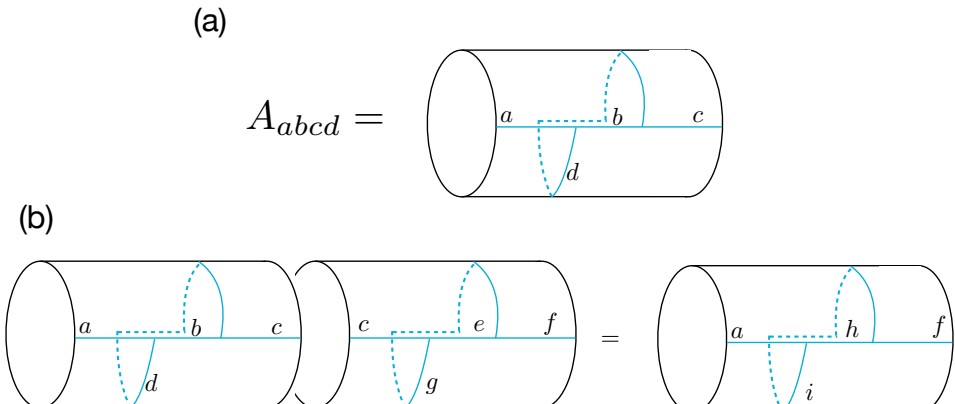

Figure 5: (a) The MPO represents the cylinder (tube). (b) The matrix interpretation of the Ocneanu's tube algebra.

The multiplication allows us to find the central idempotents,

$$\mathcal{P}_{1,1\bar{1}} = \frac{1}{\sqrt{5}\phi}\left(A_{1111} + \phi A_{1\tau1\tau}\right),$$

$$\mathcal{P}_{1,\tau\bar{\tau}} = \frac{1}{\sqrt{5}\phi}\left(\phi^2 A_{1111} - \phi A_{1\tau1\tau}\right),$$

$$\mathcal{P}_{\tau,\tau\bar{1}} = \frac{1}{\sqrt{5}\phi}\left(A_{\tau\tau\tau1} + e^{-i\cos^{-1}\frac{-\phi}{2}}A_{\tau1\tau\tau} + \sqrt{\phi}e^{i\cos^{-1}\frac{-1}{2\phi}}A_{\tau\tau\tau\tau}\right),$$

$$\mathcal{P}_{\tau,1\bar{\tau}} = \frac{1}{\sqrt{5}\phi}\left(A_{\tau\tau\tau1} + e^{i\cos^{-1}\frac{-\phi}{2}}A_{\tau1\tau\tau} + \sqrt{\phi}e^{-i\cos^{-1}\frac{-1}{2\phi}}A_{\tau\tau\tau\tau}\right),$$

$$\mathcal{P}_{\tau,\tau\bar{\tau}} = \frac{1}{\sqrt{5}\phi}\left(\phi A_{\tau\tau\tau1} + \phi A_{\tau1\tau\tau} + \frac{1}{\sqrt{\phi}}A_{\tau\tau\tau\tau}\right). \tag{A.6}$$

Here we can express $\cos^{-1}\frac{-\phi}{2} = \frac{4\pi}{5}$ and $\cos^{-1}\frac{-1}{2\phi} = \frac{3\pi}{5}$. Now let us perform the Galois conjugation, $\phi \to -1/\phi = \phi'$. The central idempotents of the Galois conjugated tube algebra of the Fibonacci string-net model (Yang-Lee model) are

$$\mathcal{P}_{1,1\bar{1}} = -\frac{1}{\sqrt{5}\phi'}\left(A_{1111} + \phi' A_{1\tau1\tau}\right),$$

$$\mathcal{P}_{1,\tau\bar{\tau}} = -\frac{1}{\sqrt{5}\phi'}\left(\phi'^2 A_{1111} - \phi' A_{1\tau1\tau}\right),$$

$$\mathcal{P}_{\tau,\tau\bar{1}} = -\frac{1}{\sqrt{5}\phi'}\left(A_{\tau\tau\tau1} + e^{-i\cos^{-1}\frac{-\phi'}{2}}A_{\tau1\tau\tau} + \sqrt{\phi'}e^{i\cos^{-1}\frac{-1}{2\phi'}}A_{\tau\tau\tau\tau}\right),$$

$$\mathcal{P}_{\tau,1\bar{\tau}} = -\frac{1}{\sqrt{5}\phi'}\left(A_{\tau\tau\tau1} + e^{i\cos^{-1}\frac{-\phi'}{2}}A_{\tau1\tau\tau} + \sqrt{\phi'}e^{-i\cos^{-1}\frac{-1}{2\phi'}}A_{\tau\tau\tau\tau}\right),$$

$$\mathcal{P}_{\tau,\tau\bar{\tau}} = -\frac{1}{\sqrt{5}\phi'}\left(\phi' A_{\tau\tau\tau1} + \phi' A_{\tau1\tau\tau} + \frac{1}{\sqrt{\phi'}}A_{\tau\tau\tau\tau}\right). \tag{A.7}$$

Here the minus signs in front of the central idempotents ensure $\mathcal{P}_i^2 = \mathcal{P}_i$, and we can express $\cos^{-1}\frac{-\phi'}{2} = \frac{2\pi}{5}$ and $\cos^{-1}\frac{-1}{2\phi'} = \frac{\pi}{5}$.

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
