# Peer review of "Relating non-Hermitian and Hermitian quantum systems at criticality"

_SciPost Physics, doi:SciPost Phys. Core 6, 062 (2023)_

## Round 2 · Referee Report · Anonymous (Referee 1) · 2023-4-22

Strengths

  1. This manuscript discusses three different relationships between Hermitian and non-Hermitian systems, which may be helpful for a unified understanding of the quantum critical behavior in non-Hermitian systems.

Weaknesses

  1. The three relationships studied in this manuscript are not found in this manuscript but were already developed in the previous works.

  2. The three relationships do not seem to be related to each other for now, which makes the main theme of this manuscript ambiguous.

  3. This manuscript contains many typographic or grammatical errors.

Report

The authors discuss three different relationships between Hermitian and non-Hermitian systems and their relevance to quantum critical behavior. The first relationship is based on a similarity transformation that preserves both energy and entanglement spectra. The second relationship is based on Naimark’s dilation, for which a non-Hermitian system is embedded in a larger Hermitian system. The third relationship is based on Galois conjugation, for which the negative central charges can be understood by the transformation between the conformal vacua.

Recently, quantum phase transitions of non-Hermitian systems have attracted considerable interest. Despite such interest, the quantum critical behavior of non-Hermitian systems, especially that characterized by negative central charges, has remained elusive. In this respect, I believe that this manuscript, which discusses the three relationships between Hermitian and non-Hermitian systems, should make a significant contribution. Thus, I would like to recommend publication of this manuscript in SciPost Physics.

Requested changes

Before the publication, I would like to request that the authors address the following concerns.

(1) In the first sentence of Sec. 2, the manuscript reads, “In a non-Hermitian system $H\neq H^{\dagger}$, the eigenstates form the biorthonormal basis with $H | \psi_{Rn} \rangle = E_n | \psi_{Rn} \rangle$, $H^{\dagger} | \psi_{Ln} \rangle = E_n^{*} | \psi_{Ln} \rangle$, $\langle \psi_{Ln} | \psi_{Rm} \rangle = \delta_{n, m}$ ”. However, this statement is not necessarily true. In fact, at an exceptional point, a right eigenstate and the corresponding left eigenstate are orthogonal to each other, i.e., $\langle \psi_{Ln} | \psi_{Rm} \rangle = 0$ even for $n=m$. Since exceptional points should underlie the quantum critical phenomena studied in this manuscript, the authors should correct this statement.

(2) In Sec. 2.1, the manuscript reads, “the spectrum is real if the quasi-Hermitian condition (??) is assumed”. The authors should correct the equation number and clarify the definition of quasi-Hermiticity.

(3) The manuscript contains many typographic or grammatical errors. Please correct them carefully.

  • validity: good
  • significance: ok
  • originality: ok
  • clarity: low
  • formatting: acceptable
  • grammar: below threshold

---

## Round 2 · Referee Report · Anonymous (Referee 2) · 2023-5-27

Strengths

1- They considered three specific models for different cases of transformation. 2- They extracted the central charge from the novel definition of entanglement entropy for non-Hermitian systems. 3- Scaling analysis of the entanglement entropy in case 2 with large system sizes.

Weaknesses

1- The system sizes in case 1 and case 3 are too small and not convincing. Case 2 may not be reproducible without detailed information on calculations about some confusing results. 2- Demonstration of these central charges from the scaling of entanglement entropy in Hermitian and non-Hermitian systems is not a ground-breaking discovery. 3- The study is a follow-up work of the author's previous publication and is not opening a new pathway or research direction. 4- The manuscript contains many typos and unclear expressions. The introduction/motivation and detailed calculations can be improved. 5- The manuscript lacks sufficient references to support and contextualize the presented research. 6- The current manuscript looks more like a brief report on the model calculations. 7- Do not have a broad interest or link to other research areas.

Report

This paper examines the relationship between Hermitian and non-Hermitian quantum systems at criticality using various transformations. They consider three different scenarios and classify them based on whether there are changes in the energy spectrum and the eigenvalue spectrum of the reduced density matrix. The authors employ a novel definition of entanglement entropy for non-Hermitian systems to extract the central charge, but its interpretation as an entanglement measure remains unclear. The paper analyzes three models, providing valuable insights into the physical properties of these systems. While the results are not groundbreaking discoveries, they contribute to the specific field. The paper is perceived as a follow-up to the author's previous publication [27], which may lower its novelty. The scaling of entanglement entropy in case 2 is excellently sized up, but concerns are raised about the small system sizes in cases 1 and 3, which may pull the rigor and validity of the results. Especially in case 3, the data shown in Fig.4 have a large deviation from the fitting curve. Similarity transformation in case 1 is probably trivially without problem, however, the authors are suggested to carefully examine whether they define the transverse field Ising model Hamiltonian in terms of spin-1/2 operators or merely Pauli matrices, potentially leading to different values of the quantum critical point. Numerous typos and poorly written sections detract from the manuscript's quality. Additionally, it may not have broad interest or relevance to other research areas. For example, Naimark's dilation method does not have a solution in general Hamiltonians. While the manuscript does not currently meet the criteria of Scipost Physics, solely showing $S_A$ it looks more like a brief report without enough introduction about the background and the motivation. I recommend the manuscript be transferred to SciPost Physics Core after careful revision.

I kindly ask the authors to please take a look at the detailed comments, questions, and suggestions listed below. I hope these will be helpful in improving the manuscript.

Requested changes

1- Instead of using "physical spectrum", it would be more appropriate to use "energy spectrum" to refer to the set of energies of a physical system.

2- What the "entanglement spectrum" refers to in the manuscript is unclear. The well-known definition in the Hermitian system is the eigenvalues of the entanglement Hamiltonian $H_E$, where the reduced density matrix can be written as a thermal mixed state with unity temperature $\rho_A=e^{-H_E}$ [Phys. Rev. Lett. 101, 010504 (2008)]. Only a few researchers refer to the eigenvalues of the reduced density matrix as the entanglement spectrum, e.g. [Phys. Rev. A 78, 032329 (2008)], while the majority still consider it as such only after taking the logarithm. Since the authors adopt a novel definition of the entanglement entropy for non-Hermitian systems with biorthogonal density matrix [41], it would be helpful if the authors clarify whether the "entanglement spectrum" in their study pertains to just the eigenvalues $\omega_i$ of the reduced density matrix, or $\xi_i=-\ln\omega_i$, or $\ln|\omega_i|$, or something else. There is also a generalization to the non-Hermitian entanglement spectrum. [Phys. Rev. A 99, 052118 (2019)]

3- For all 3 cases they discussed, both the energy and entanglement spectra must be shown for better viewing of the changes before and after the transformations.

4-Previous studies have explored the entanglement spectrum in 1D Hermitian systems, which has been described by a universal scaling function that depends solely on the central charge of the underlying conformal field theory [Phys. Rev. A 78, 032329 (2008)]. If the authors intend to discuss the entanglement spectrum, it would be helpful to include a description of the relevant phenomena.

5- The introduction section appears too condensed. It would be helpful to provide more information regarding the non-Hermitian skin effect, as it seems to occur in the coupled Hatano-Nelson model presented in Case 2 if the open boundary condition is applied.

6- Non-Hermitian systems can exhibit both quantum critical points (QCP) and exceptional points (EP). It would be beneficial to discuss both concepts and clarify their differences. Furthermore, providing evidence for the investigated models to determine whether the QCP is an EP or just a QCP would enhance the validity of the analysis. Possibly a convenient way is to consider the fidelity susceptibility. [Phys. Rev. Res. 3, 013015 (2021); Quantum 7, 960 (2023)]

7- In case 2, I found two EPs at $k=\pm\frac{\pi}{2}$. I am confused whether the authors assume a half-filled ground state, with two particles occupying the momentum EPs. Please provide further explanation on this issue. By the way, the current expressions of the left/right eigenvectors in Eq.(12),(13) blow up at $|k|=\frac{\pi}{2}$.

8- Please provide detailed calculations of the entanglement entropy of dilated ground state from Eq.(14). If the authors apply the same technique of correlation matrix method as used in Ref.[27,41], please include the original reference [J. Phys. A: Math. Gen. 36, L205 (2003)] for the Hermitian system, and [SciPost Phys. 7, 069 (2019)] for the extension to the non-Hermitian system.

9- At the end of Sec. 2.2, the authors claim their results agree with the partition function analysis. Please provide the references for the partition function analysis on the model.

10- Background information about the Yang-Lee edge singularity is necessarily written in the introduction section. If the non-Hermitian Ising model Eq.(2) in case 1 is not related to the Yang-Lee edge singularity, the authors are suggested to clarify and put forward the motivation or the significance of the model Eq.(2). Otherwise, it currently looks like a good exercise of similarity transformation.

11- More introduction on conformal field theory (CFT) is requested, including its connection to the theory of critical phenomena and the study of quantum criticality. It may be helpful to briefly explain the concept of the scaling of $S_A$ and how it relates to CFT. Regarding non-unitary CFT, it would be useful to discuss its applications in the study of non-Hermitian systems and how it differs from unitary CFT. A brief explanation of bc-ghost CFT and its significance in string theory could also be included. Additionally, it may be worth mentioning why the central charge and entanglement entropy are related, as this is the main focus of this paper.

12- The scaling property of the newly defined entanglement entropy appears to be still an open question. However, the manuscript should explicitly state the scaling formula used and address the boundary conditions for all considered models. i.e. c/3 or c/6 is used? [J. Stat. Mech. (2004) P06002]

13- More introduction about the new definition of non-Hermitian entanglement entropy is necessary, including how to experimentally measure the expectation value of an observable in the left/right eigenstate basis. $\langle O\rangle_n=\langle\psi_{Ln}|O|\psi_{Rn}\rangle$

14- The notation used for the new entropy definition is unclear and potentially misleading. Usually, $|\rho_A|$ represents the determinant of the matrix $\rho_A$, not the absolute value. To properly express the entropy, the following expression can be used: $S_A = -\sum_i \omega_i \ln|\omega_i|$, where $\omega_i$ denotes the $i$th eigenvalue of the reduced density matrix $\rho_A$, and $|\omega_i|$ takes the absolute value for the complex eigenvalues $\omega_i$. Additionally, it would be helpful to clarify the physical meaning if it is complex.

15- The authors have also mentioned the Renyi entropy in Eq.(1), but no related data has been presented. If the Renyi entropy does not provide any additional information, my suggestion would be to consider omitting it from the paper. Alternatively, the authors could provide some data to support its inclusion.

16- In Eq.(2), whether $S_i^z$ is the spin-1/2 operator or merely a Pauli matrix should be clearly stated, and please check whether $J/\beta=1$ is the critical point. In Fig.1, the total system size $L$ should be explicitly written. Below Eq.(3), $\gamma^\mp$ is a confusing notation and should be replaced by $\gamma^{\mp1}$ or $\gamma$ and $\frac{1}{\gamma}$. Below Eq.(2), the complex numbers $\epsilon_1\neq\epsilon_2^*$ is the condition for the non-Hermitian Ising model. Eq.(4),(8) have missing parentheses for the summation.

17- Most calculations for case 1 following Ref.[42], which employed the concept of metric in the Hilbert space. If the related concept has been used in the manuscript, the authors should address and include relevant references.

18- Page 4, line 4, quasi-Hermitian condition (??), please amend. Page 9, line 2, "discussed". Page 10, sec. A.1, line 6, "quasiparticle". Line 8, before Eq.(22), "state".

19- The scaling of $S_A$ with fixed ratio $\frac{L_A}{L}=\frac{1}{2}$ for case 1 and case 3 are requested, similar to Fig.2(b). For case 2, the current presentation of Fig.2(b) makes the data look like horizontal lines. I suggest reorganizing by showing the c=2 [Fig.2(b)]and c=-4 [Fig.2(c)] separately. By the way, obtaining the central charge of c=-4 for the non-interacting fermion ladder is not a novel result since it has already been reported in their previous work [41].

20- It is suggested to briefly discuss the background information for each model (non-Hermitian Ising chain, Hatano-Nelson ladder, and the Fibonacci anyon chain) before presenting the data. What is the significance of these models? What makes them representative? Why did the authors choose these models instead of others?

21- The authors could clarify that their choice of using Naimark's dilation method to map the non-Hermitian Hamiltonians to Hermitian is specific to the models studied in this paper, as well as the specific case of the Fibonacci anyon chain. There exist other methods for mapping non-Hermitian Hamiltonians into Hermitian that could be used and may lead to different outcomes depending on the specific system.

22- The total size $L=16$ should be explicitly mentioned in the caption of Fig.4. Although the results in Fig4(a) and 4(b) claiming c=-3/5 and c=-22/5 are intriguing, the displayed data deviates significantly from the fitting curve, especially when one of the subsystem sizes is small, possibly due to the small total system size. In order to obtain more reliable data for their interesting claims, it is suggested to increase the size of the system in the PEPS method and fix the ratio $\frac{L_A}{L}=\frac{1}{2}$ for scaling.

23- The authors are requested to write a paragraph in the appendix explaining how they obtain the first excited state by using PEPS.

---

## Round 3 · Referee Report · Anonymous · 2023-8-2

Report

I would very much appreciate the response and the corresponding revision of the manuscript. The authors have addressed all of my concerns satisfactorily and revised the manuscript accordingly. I would like to recommend publication of this manuscript in SciPost Physics.

---

## Round 3 · Referee Report · Anonymous · 2023-8-4

Report

This revision of the manuscript contains some improvements, but there are still some minor typos that haven't been corrected. Despite this, the overall article doesn't present any groundbreaking insights. The data presented in Case 1 and Case 2 are convincing; however, Case 3 still presents challenges in obtaining rigorous numerical results, resulting in ambiguous data. In my opinion, if the authors were to employ a sophisticated numerical method like PEPS, they could potentially present numerical results for a system size much larger than what exact diagonalization can achieve.

This submission does not meet the criteria of SciPost Physics, but does meet those of SciPost Physics Core, where it could be published.

Requested changes

Please carefully proofread the whole manuscript and make sure to reduce the possible misleading expression and typos. e.g. a matrix determinant $|A|$, missing parentheses in Eq.(5), Pauli matrix or Spin-1/2 matrix?, note that $\mathbf{S}=\frac{1}{2}\mathbf{\sigma}$, $S^{\pm}=S^x\pm iS^y=\frac{1}{2}(\sigma^x\pm i\sigma^y)$. The matric operator $g=\sum_n|\psi_{Ln}\rangle\langle\psi_{Ln}|$. Eigenvalue degeneracy does not affect biorthogonality. Fig.2(b): what is the entanglement spectrum refer to? It looks like $\epsilon_E=-\ln\omega$? It is inconsistent with the author's reply and the revised manuscript: they refer to the eigenvalues of the reduced density matrix $\omega$. The above are just what I found at a glance. It is the author's responsibility to check through their manuscript.

---

## Round 3 · Author Response

A. Response to first referee
We thank the first referee for his/her careful review of our manuscript. The first referee commented that our paper ”The paper analyzes three models, providing valuable insights into the physical properties of these systems”. We have followed the first referee’s suggestion and rewritten our manuscript accordingly.

1\. Instead of using ”physical spectrum”, it would be more appropriate to use ”energy spectrum” to refer to the set of energies of a physical system.

Response: We have changed the ”physical spectrum” to the ”energy spectrum” in the revised manuscript.

2\. What the ”entanglement spectrum” refers to in the manuscript is unclear. The well-known definition in the Hermitian system is the eigenvalues of the entanglement Hamiltonian HE, where the reduced density matrix can be written as a thermal mixed state with unity temperature $\rho_A = e^{−H_E}$ [Phys. Rev. Lett. 101, 010504 (2008)]. Only a few researchers refer to the eigenvalues of the reduced density matrix as the en- tanglement spectrum, e.g. [Phys. Rev. A 78, 032329 (2008)], while the majority still consider it as such only after taking the logarithm. Since the authors adopt a novel definition of the entanglement entropy for non-Hermitian systems with biorthogonal density matrix [41], it would be helpful if the authors clarify whether the ”entanglement spectrum” in their study pertains to just the eigenvalues ωi of the reduced density matrix, or $zeta_i = − \ln \omega_i$, or $\ ln |\omega_i|$, or something else. There is also a generalization to the non-Hermitian entanglement spectrum. [Phys. Rev. A 99, 052118 (2019)].

Response: We use the eigenvalues of the reduced density matrix as the entanglement spectrum. We have added our definition of the entanglement spectrum in Sec.2 and in Eq. (1). Additionally, we have included plots of the entanglement spectra in Figs. 1, 2, and 4.

3\. For all 3 cases they discussed, both the energy and entanglement spectra must be shown for better viewing of the changes before and after the transformations.

Response: We have added the entanglement spectra for all the cases discussed in the manuscript. However, we have not included the energy spectra as they are not the focus of our study.

4\. Previous studies have explored the entanglement spectrum in 1D Hermitian systems, which has been described by a universal scaling function that depends solely on the central charge of the underlying conformal field theory [Phys. Rev. A 78, 032329 (2008)]. If the authors intend to discuss the entanglement spectrum, it wouldbe helpful to include a description of the relevant phenomena.
Response: We have added a brief discussion about the entanglement spectrum for 1D Hermitian critical systems and cited the paper accordingly.

5\. The introduction section appears too condensed. It would be helpful to provide more information regarding the non-Hermitian skin effect, as it seems to occur in the coupled Hatano-Nelson model presented in Case 2 if the open boundary condition is applied.

Response: In our study, the non-Hermitian skin effect is irrelevant. Additionally, the coupled Hatano-Nelson model has real energy spectrum. The criterion for the non-Hermitian skin effect requires a complex energy spectrum winding around the origin. Therefore, there is no non-Hermitian skin effect in the coupled Hatano-Nelson model.

6\. Non-Hermitian systems can exhibit both quantum critical points (QCP) and exceptional points (EP). It would be beneficial to discuss both concepts and clarify their differences. Furthermore, providing evidence for the investigated models to determine whether the QCP is an EP or just a QCP would enhance the validity of the analysis. Possibly a convenient way is to consider the fidelity susceptibility. [Phys. Rev. Res. 3, 013015 (2021); Quantum 7, 960 (2023)]

Response: We have added a short discussion about the differences between the QCP and the EP, and cited the papers accordingly.

7\. In case 2, I found two EPs at $k = \pm \pi/2$ . I am confused whether the authors assume a half-filled ground state, with two particles occupying the momentum EPs. Please provide further explanation on this issue. By the way, the current expressions of the left/right eigenvectors in Eq.(12),(13) blow up at $|k| = \pi/2$ .

Response: We thank the referee for this comment. Indeed, the singularity of the left/right eigenvector occurs at $|k| = \pi/2$ . To avoid this singularity, we compute the entanglement entropy near the transition point (quantum critical point), by using a small momentum shift $\delta = 0.000001$. We have included a detailed explanation of this approach in the revised manuscript.

8\. Please provide detailed calculations of the entanglement entropy of dilated ground state from Eq.(14). If the authors apply the same technique of correlation matrix method as used in Ref.[27,41], please include the original reference [J. Phys. A: Math. Gen. 36, L205 (2003)] for the Hermitian system, and [SciPost Phys. 7, 069 (2019)] for the extension to the non-Hermitian system.

Response: We have added detailed calculations of the entanglement entropy of Eq. (14) by using the correlation matrix method and cited the papers accordingly.

9\. At the end of Sec. 2.2, the authors claim their results agree with the partition function analysis. Please provide the references for the partition function analysis on the model.

Response: We have added the reference accordingly.

10\. Background information about the Yang-Lee edge singularity is necessarily written in the introduction section. If the non-Hermitian Ising model Eq.(2) in case 1 is not related to the Yang-Lee edge singularity, the authors are suggested to clarify and put forward the motivation or the significance of the model Eq.(2). Otherwise, it currently looks like a good exercise of similarity transformation.

Response: We have added a short introduction of the Yang-Lee edge singularity in Sec. 2.1. However, we would like to clarify that the non-Hermitian Ising model serves as a valuable exercise in the context of the similarity transformation but is not directly related to the Yang-Lee edge singularity.

11\. More introduction on conformal field theory (CFT) is requested, including its connection to the theory of critical phenomena and the study of quantum criticality. It may be helpful to briefly explain the concept of the scaling of SA and how it relates to CFT. Regarding non-unitary CFT, it would be useful to discuss its applications in the study of non-Hermitian systems and how it differs from unitary CFT. A brief explanation of bc-ghost CFT and its significance in string theory could also be included. Additionally, it may be worth mentioning why the central charge and entanglement entropy are related, as this is the main focus of this paper.

Response: We have added a new subsection 2.1 introducing the entanglement entropy scaling and the CFTs, along with an explanation of how the central charges can be extracted from the entanglement entropy scaling. Additionally, we have included a paragraph discussing non-unitary CFTs and bc-ghost CFTs.

12\. The scaling property of the newly defined entanglement entropy appears to be still an open question. However, the manuscript should explicitly state the scaling formula used and address the boundary conditions for all considered models. i.e. $c/3$ or$ c/6$ is used? [J. Stat. Mech. (2004) P06002]

Response: We only consider the periodic boundary condition in our cases, for which the $c/3$ is used. We add the discussion about the boundary condition in the revised manuscript.

13\. More introduction about the new definition of non-Hermitian entanglement entropy is necessary, including how to experimentally measure the expectation value of an observable in the left/right eigenstate basis. $\langle O \rangle_n = \langle \psi_{Ln}|O| \psi_{Rn} \rangle$.

Response: We have provided a more comprehensive introduction to the generic entanglement entropy for the non-Hermitian systems and discussed the potential experimental measures by the bi-orthonormal basis.

14\. The notation used for the new entropy definition is unclear and potentially misleading. Usually, $\rho_A$ represents the determinant of the matrix $\rho_A$, not the absolute value. To properly express the entropy, the following expression can be used: $S_A=\sum_i \omega_i \ln |\omega_i|$, where $\omega_i$ denotes the i-th eigenvalue of the reduced density matrix $\rho_A$, and $|\omega_i|$ takes the absolute value for the complex eigenvalues $\omega_i$. Additionally, it would be helpful to clarify the physical meaning if it is complex.

Response: We have expressed Eq. (1) as $S_A=\sum_i \omega_i \ln |\omega_i|$.

15\. The authors have also mentioned the Renyi entropy in Eq.(1), but no related data has been presented. If the Renyi entropy does not provide any additional information, my suggestion would be to consider omitting it from the paper. Alternatively, the authors could provide some data to support its inclusion.

Response: We have removed the definition of Renyi entropy in Eq. (1).

16\. In Eq.(2), whether$ S^i_z$ is the spin-$1/2$ operator or merely a Pauli matrix should be clearly stated, and please check whether $J/\beta = 1$ is the critical point. In Fig.1, the total system size L should be explicitly written. Below Eq.(3), $\gamma^\pm$ is a confusing notation and should be replaced by $\gamma^{\pm 1}$ or $\gamma$ and $1/\gamma$. Below Eq.(2), the complex numbers $\epsilon_1 \neq \epsilon_2^*$ is the condition for the non-Hermitian Ising model. Eq.(4),(8) have missing parentheses for the summation.

Response: We thank the referee for pointing out our typos. We have changes $S^\alpha$ to the Pauli matrices $\sigma^\alpha$.$J/\beta = 1$ is the critical point. We have explicitly written down the total systems size $L = 16$. $\gamma^\pm$ changes to $\gamma^{\pm 1}$ and the missing parentheses of the summations are adding back.

17\. Most calculations for case 1 following Ref.[42], which employed the concept of metric in the Hilbert space. If the related concept has been used in the manuscript, the authors should address and include relevant references.

Response: The concept of metric in the Hilbert space is similar to the bi-orthogonal formalism. We have added a short discussion about the metric in the revised manuscript.

18\. Page 4, line 4, quasi-Hermitian condition (??), please amend. Page 9, line 2, ”discussed”. Page 10, sec. A.1, line 6, ”quasiparticle”. Line 8, before Eq.(22), ”state”.

Response: We thank the referee for pointing out our typos. We have corrected the typos accordingly.

19\. The scaling of $S_A$ with fixed ratio $L_A/L = 1/2$ for case 1 and case 3 are requested, similar to Fig.2(b). For case 2, the current presentation of Fig.2(b) makes the data look like horizontal lines. I suggest reorganizing by showing the $c = 2$ [Fig.2(b)]and $c = −4$ [Fig.2(c)] separately. By the way, obtaining the central charge of$ c = −4$ for the non-interacting fermion ladder is not a novel result since it has already been reported in their previous work [41].

Response: We have included the scaling of $S_A$ with fixed ratio $L_A/L = 1/2$ for case 1. As for case 3, it is challenging to numerically compute for large sizes, and when $L_A/ L= 1/2$ for small sizes, it may not yield an accurate central charge. However, we believe that our demonstration of the entanglement entropy scaling for small sizes in the case 3 sufficiently elaborates on our idea. We have separated the $c = 2$ and $c = −4$ to Fig. 2(d) and 2(c), respectively. The central charge of $c = −4$ for the non-interacting fermion ladder is not the model studied in our previous work [41].

20\. It is suggested to briefly discuss the background information for each model (non-Hermitian Ising chain, Hatano-Nelson ladder, and the Fibonacci anyon chain) before presenting the data. What is the significance of these models? What makes them representative? Why did the authors choose these models instead of others?

Response: Our study focuses on three types of transformations relating non-Hermitian and Hermitian systems at criticality. Regarding the first and the second transformations, there is no specific reason for choosing the non-Hermitian Ising chain and the Hatano-Nelson ladder, respectively, except for their suitability in illustrating the transformations. However, it is worth noting that, to the best of our knowledge, the simplest model for realizing the Galois conjugation is the Fibonacci anyon chain, as we have discussed in the context of the third transformation.

21\. The authors could clarify that their choice of using Naimark’s dilation method to map the non-Hermitian Hamiltonians to Hermitian is specific to the models studied in this paper, as well as the specific case of the Fibonacci anyon chain. There exist other methods for mapping non-Hermitian Hamiltonians into Hermitian that could be used and may lead to different outcomes depending on the specific system.

Response: The reason for the choice of Naimark’s dilation method is as follows: In Ref. [L.-M. Chen, S. A. Chen and P. Ye, Entanglement, Non-Hermiticity, and Duality, SciPost Phys. 11, 3 (2021)], the dual transformation can only map the non-Hermitian systems to the other non-Hermitian systems which they referred to as type-II models. The transformed Hermitian models of these type-II models cannot be constructed by the similarity transformation. Hence we consider the Naimark’s dilation method, which preserves the energy spectrum. As for the third transformation, which does not preserve both the energy and the entanglement spectra, to the best of our knowledge, the only known example is the Galois conjugation on the anyon chain. Consequently, we only investigate the Naimark?s dilation method for the second transformation and the Galois conjugations for the third transformation.

22\. The total size $L = 16$ should be explicitly mentioned in the caption of Fig.4.
Although the results in Fig4(a) and 4(b) claiming $c = −3/5$ and $c = −22/5$ are
intriguing, the displayed data deviates significantly from the fitting curve, especially
when one of the subsystem sizes is small, possibly due to the small total system size.
In order to obtain more reliable data for their interesting claims, it is suggested to in-
crease the size of the system in the PEPS method and fix the ratio $L_A/L = 1/2$ for scaling.

Response: We have explicitly mentioned $L = 16$ in Fig. 4. Using PEPS method for case 3 is not the focus of our study.

23\. The authors are requested to write a paragraph in the appendix explaining how they obtain the first excited state by using PEPS.

Response: The PEPS is introduced in the case 3 because the matrix product operator (MPO) algebra of string-net PEPS can be explicitly written down. The sectors after the Galois conjugation can be obtained analytically, and we can show that the vacuum sector of the Galois conjugated state has the conformal weight $h = 0$ which is NOT the ground state. This is because there is another state with negative conformal weight $h = −1/5$, which will be the ground state of the Galois-conjugated model. We have added a short discussion in the revised manuscript.

B. Response to second referee
We thank the second referee for his/her careful review of our manuscript. The second referee commented that our paper ” I would like to recommend publication of this manuscript in SciPost Physics”. We have followed the second referee’s suggestion and rewritten our manuscript accordingly.

1\. In the first sentence of Sec. 2, the manuscript reads, "In a non-Hermitian system $H \neq H^\dagger$, the eigenstates form the biorthonormal basis with $H | \psi_{Rn} \rangle= E_n | \psi_{Rn} \rangle$, $H^\dagger | \psi_{Ln} \rangle= E_n^* | \psi_{Ln} \rangle$, $\langle \psi_{Ln} | \psi_{Rm}\rangle=\delta_{n,m}$". However, this statement is not necessarily true. In fact, at an exceptional point, a right eigenstate and the corresponding left eigenstate are orthogonal to each other, i.e., $\langle \psi_{Ln}|\psi_{Rm}\rangle = 0$ even for $n = m$. Since exceptional points should underlie the quantum critical phenomena studied in this manuscript, the authors should correct this statement.

Response: We thank the referee for the comment. We have corrected the statement and clarified that when there is an exceptional point, the bi-orthonormal basis is not complete.

2\. In Sec. 2.1, the manuscript reads, ”the spectrum is real if the quasi-Hermitian condition (??) is assumed”. The authors should correct the equation number and clarify the definition of quasi-Hermiticity.

Response: We have corrected the typos accordingly, and written down the definition of quasi- Hermiticity.

3\. The manuscript contains many typographic or grammatical errors. Please correct them carefully.

Response: We thank the referee carefully read our manuscript. We have corrected the typo- graphic or grammatical errors in the revised manuscript.

---

## Round 3 · List of Changes

We made the following changes to the text:

1. We replace ”physical spectrum” with ”energy spectrum”.
2. We discuss the universal scaling property of the entanglement entropy in critical systems and add section 2.1 for a detail discussion.
3. Wediscussthemetricoperator,themeasurementpostulate,andthebi-othonormalformalism in section 2.
4. We express the generic entanglement entropy [Eq. (1)] by the eigenvalues of the reduced density matrix, which refers to the entanglement spectrum.
5. We remove the Renyi entropy in section 2.
6. We add a short introduction to the non-unitary CFTs, including the bc-ghost CFT in section 2.1.
7. We add detailed calculations of the entanglement entropy for the second case by introducing a momentum shift.
8. We change Sα to σα for the non-Hermitian Ising model.
9. We define the quasi-Hermiticity for the non-Hermitian Ising model.
10. We add the parentheses for the summations in Eqs. (5) and (9).
11. We add numerical calculations of the entanglement entropy of the non-Hermitian Ising model for large sizes.
12. We add the entanglement spectra for all the cases.
13. We add the discussion for the entanglement entropy calculations for the free fermion cases by using the correlation matrix method and cite the references accordingly.
14. We add the discussion about the vacuum sector for the Galois conjugated anyon chain being the first excited state in section 2.4.
15. We correct several typos and grammatical errors.

We also substantially remade Figs. 1-4, as follows:
1. Fig. 1: We add (a) for the entanglement spectra and (c) for the half-size entanglement entropy scaling for the large size calculation.
2. Fig. 2: We add (b) for the single-particle entanglement spectra, and separate the entangle- ment entropy scalings for c = −4 and c = 2 into (c) and (d).
3. Fig. 4: We add (a) and (b) for the entanglement spectra.

---

## Editorial Decision

published